# The Relationship between Types of Os Trigonum and Findings of Conventional Ankle Magnetic Resonance Imaging: A Study Based on Three-Dimensional Magnetic Resonance Imaging

**DOI:** 10.3390/diagnostics14030283

**Published:** 2024-01-28

**Authors:** Kyu-Hong Lee, Ro-Woon Lee, Yeo-Ju Kim

**Affiliations:** 1Department of Radiology, College of Medicine, Inha University, Incheon 22332, Republic of Korea; 2Department of Radiology, College of Medicine, Hanyang University, Seoul 04763, Republic of Korea

**Keywords:** os trigonum, magnetic resonance imaging (MRI), ankle pain

## Abstract

This study aimed to investigate the dimensions and types of the os trigonum and evaluate their relationship with various pathologic conditions on the posterior ankle using ankle MRI images. A total of 124 non-contrast-enhanced ankle and foot MR images of 123 consecutive patients were included in this retrospective study. The images were presented randomly, and they contained no patient information. The MR images were retrospectively and independently reviewed by two reviewers with a fellowship-trained musculoskeletal radiologist. The images were classified as type I and II based on the ossicle’s medial border overlying the talus’s posterior process and the groove for the flexor hallucis longus tendon (FHL). The study revealed that patients with type II os trigonum had a longer transverse diameter of the ossicle than type I, and there were statistically significant differences. Detachment status tended to be less in type I than in type II os trigonum, and the differences between the groups were statistically significant. There were no significant differences between type I and II os trigonum regarding posterior talofibular ligament (PTFL) abnormality, bone marrow edema, FHL tenosynovitis, and posterior synovitis. The study concluded that the os trigonum is a common cause of posterior ankle impingement, and type II os trigonum has a longer transverse diameter of the ossicle than type I.

## 1. Introduction

Posterior ankle impingement (PAI) is a common cause of posterior ankle pain during plantar flexion. It is caused by overuse, repetitive plantarflexion, or a traumatic event, often in combination with congenital anatomic anomalies [1]. Moreover, the essential cause of PAI is the presence of an os trigonum. It is a separate ossicle located posterior to the lateral tubercle of the posterior process of the talus (Stieda’s process), just lateral to the flexor hallucis longus groove [2]. It represents a failure of fusion of the lateral tubercle of the posterior process.

Rosenmuller first described the os trigonum as an accessory ossicle [3]. The name originated from the trigonal shape of its three surfaces: the anterior surface adjacent to the talus, the posterior surface with its ligamentous attachment to the posterior talofibular ligament (PTFL), and the inferior surface adjacent to the calcaneus. The relationship between posterior ankle pain and the radiological findings of an os trigonum was first described by McDougall [4]. During plantarflexion, the os trigonum and surrounding soft tissue become impinged between the distal tibia’s posterior aspect and the calcaneus’s superoposterior surface. Also, flexor hallucis longus (FHL) tenosynovitis, PTFL abnormality, synovitis on the posterior talocalcaneal joint, and bone marrow edema at the os trigonum often coexist with a PAI because of their anatomical proximity. For the same reason above, the os trigonum is considered one of the most common causes of posterior ankle impingement.

Since the first descriptions of the os trigonum, a wide variation in occurrence (1.7–12.7%) has been reported [5]. However, there has yet to be a report about its characteristics, including dimensions and type of os trigonum and the relationship between the os trigonum and several pathologic conditions on the posterior ankle.

This study aimed to examine the dimension and type of the os trigonum to evaluate the relationship between dimension and type, degree of PTFL attachment to the os trigonum, abnormality of PTFL, tenosynovitis of the FHL tendon, bone marrow edema at the os trigonum, the detachment status of the os trigonum from the talus, and synovitis on the posterior talocalcaneal joint using ankle MR images.

## 2. Materials and Methods

### 2.1. Study Population

This retrospective study was approved by the institutional review board of Inha University Hospital, which waived the requirement for informed patient consent (IRB No. 2020-02-013). Preoperative ankle and foot MR images of consecutive patients who underwent open or arthroscopic surgery in our institution between August 2011 and July 2019 and were diagnosed as os trigonum by the operation were evaluated in this study. A total of 142 patients who underwent arthroscopic and open surgery for the removal of the os trigonum were included. Patients with a history of infection (*n* = 3), inflammatory arthropathy (*n* = 3), gout (*n* = 2), Achilles tendon rupture (*n* = 6), and trauma (*n* = 4) were excluded (*n* = 18). A total of 124 non-contrast-enhanced ankle and foot MR images of 123 consecutive patients (male, 82; female, 41; mean age, 35.2 years; age range, 15–72 years) were included in this study. Table 1 describes the patient demographics.

### 2.2. Magnetic Resonance Imaging

Magnetic resonance imaging (MRI) was performed using 1.0 Tesla (T) Signa HDxT (GE Healthcare, WI, USA) and a 3.0 T SIGNA Architect MR scanner (GE Healthcare, SC, USA) with a dedicated coil in our hospital (*n* = 112) and different MR scanners including Echelon (Hitachi, Tokyo, Japan), Skyra, Symphony (Siemens Healthcare, Erlangen, Germany), Gyroscan Intera (Philips, MA, USA), and SIGNA Excite, HDxt, Genesis (GE Healthcare, SC, USA) in an outside hospital (*n* = 12). MR scanners in the outside hospitals operated at either 1.5 or 3.0 T magnetic field strength. The ankle and foot MRI imaging protocol included nonfat-suppressed T1-weighted fast spin echo (FSE) and T2-weighted fat suppression (FS) or proton density (PD) FS sequences for all patients, and all studies either performed in our hospital or in outside centers included sagittal and coronal T2 FS and FSE images.

### 2.3. Interpretation of Image Findings

Two reviewers retrospectively and independently reviewed all MR images with a fellowship-trained musculoskeletal radiologist (reviewer 1, Y.J.K., eighteen years of experience, and reviewer 2, R.W.L., eight years of experience in musculoskeletal MRI). These reviewers were blinded to clinical information and radiology reports. The MR images were presented in a random order, and they contained no patient information.

First, the images acquired using each MR sequence were reviewed by reader 1, and dimensions of the os trigonum were recorded, including anterior–posterior (AP) and transverse diameter on an axial T2-weighted or PD FS image, and the size of the accessory facet on a sagittal T2-weighted FS or PD FS image. Furthermore, the count of the number of ossicles was recorded by the same reader. Second, two reviewers classified all cases of os trigonum into two different types: type I and type II. Type I os trigonum is represented as the medial border of the ossicle not overlying the posterior process of the talus and an intact groove for the tendon of flexor hallucis longus (FHL) in cases of os trigonum. In cases of os trigonum where the medial border of the ossicle invading groove for the FHL tendon was seen, we considered the cases type II (Figure 1).

For the qualitative analysis, the two readers recorded about six imaging findings hypothesized to be correlated with dimensions or types of os trigonum: the extent of the attachment of posterior talofibular (PTFL) ligament, PTFL abnormality, the detachment of the os trigonum, the degree of bone marrow edema of the os trigonum, the severity of tenosynovitis of the flexor hallucis longus tendon, and synovitis in the posterior gutter. The extent of the attachment of the posterior talofibular ligament was divided into the following categories: if the PTFL was not attached to the os trigonum (grade 0), if less than half of the PTFL was attached to the os trigonum (grade 1), or if more than 1/2 portion of the PTFL was attached to the os trigonum, including in the case of the entire PTFL being attached to the os trigonum (grade 2). Abnormality of the PTLF was categorized as no visible abnormal finding (grade 0), less than half of the PTFL attached to the os trigonum (grade 1), or more than half of the PTFL attached to the ossicle (grade 2). Flexor hallucis longus (FHL) tenosynovitis was assessed by its severity and categorized as no tenosynovitis (grade 0), mild tenosynovitis (small fluid collection around the FHL tendon; grade 1), or severe tenosynovitis (a considerable amount of fluid collection extending superiorly and inferiorly into FHL tendon sheath or fluid collection with multiple synovial cysts; grade 2). Bone marrow edema (BME) in the os trigonum was assessed by the extent of BME in the os trigonum and categorized as no visible BME (grade 0), BME occupying less than half of the os trigonum (grade 1), or more than half of the BME was noted in ossicle (grade 2). The detachment state of the os trigonum was classified according to the anatomical relationship between the ossicle and posterior process of the talus; the categories included no detachment of os trigonum from the posterior process of the talus (grade 0), partial detachment, in which some cortical bridging or continuation from the os trigonum to talus was seen (grade 1), or complete detachment, wherein no demonstrable bone bridging or continuation from the os trigonum to talus was seen (grade 2). Synovitis in the posterior tibiotalar joint was classified according to its severity; the categories included no synovitis (grade 0), mild synovitis (grade 2), or severe synovitis, with a large amount of joint effusion with debris and the presence of synovial cysts (grade 3). Finally, at the end of the individual sessions, final reports for types of os trigonum and the grades of the six image findings on a three-point grading scale were made by consensus.

### 2.4. Statistical Analysis

Statistical analyses were performed using SPSS software ver. 22.0 (IBM, Armonk, NY, USA) and MedCalc for Windows, version 19.1.3 (MedCalc Software, Ostend, Belgium). The association of the type of os trigonum with age was assessed with the independent two-sample t-test. Patients with type I and type II os trigonum were compared in terms of sex and count of ossicles using a Chi-squared test and Fisher’s exact test. The correlation between the type of os trigonum and each dimension of the ossicle was assessed with the Mann–Whitney U test. This choice was guided by the distribution characteristics of our data. Preliminary analyses, including the Shapiro–Wilk test, indicated that the dimensional measurements of the os trigonum did not conform to a normal distribution. Various statistically significant associations between the type of os trigonum and six radiologic findings (PTFL attachment and abnormality, BME, detachment, FHL tenosynovitis, posterior synovitis) on ankle MRIs were evaluated with a Chi-squared test.

A value of *p * <  0.05 was considered statistically significant. Inter-observer agreements were calculated using the Kappa coefficients, which were interpreted according to the guidelines of Landis and Koch (strength of agreement for the Kappa coefficient: <0, poor; 0.01–0.2, slight; 0.21–0.4, fair; 0.41–0.6, moderate; 0.61–0.8, substantial; and 0.81–0.99, almost perfect) [6] and the percentage of time results for the same patient were concordant.

## 3. Results

### 3.1. Patient Demographics and Type

Table 1 summarizes the patient demographics and types of os trigonum of the 124 cases of 123 patients who had os trigonum and underwent surgery. These cases represented a prevalence of 29.0% (36/124) as type I and 71.0% (88/124) as type II os trigonum. The mean patient age ± standard deviation (SD) was 35.2 ± 14.6 years (age range, 15–72 years), with 87 of 123 patients (70.7%) being younger than 40 years old. The mean patient age was 30.3 years (±10.77) with a type I os trigonum and 37.1 (±15.5) with a type II os trigonum, and there were statistically significant differences in the mean age between type I and type II os trigonum. In cases of patients who visited our institution, os trigonum was more frequently found in male patients than in female patients for both type I and II (75.0% and 25.0% in type I vs. 64.3% and 35.7% in type II), and there was no statistically significant difference between the types of os trigonum. A count of one ossicle was more frequently visible in type I os trigonum cases than type II (97.2% vs. 94.3%), but there were no significant statistical differences.

### 3.2. Magnetic Resonance Imaging Findings

#### 3.2.1. Dimensions

The patients with type II os trigonum had longer transverse diameters of the ossicle (mean value ± SD: 1.21 cm ± 0.41) than type I (0.92 ± 0.36 cm), and there was a significant statistical difference (*p* = 0.001). There was insufficient evidence to show a difference in the AP diameter of the ossicle (0.83 cm ± 0.19 cm vs. 0.86 ± 0.26 cm; *p* = 0.387) or the size of the accessory facet (0.41 cm ± 0.31 vs. 0.43 cm ± 0.35, *p* = 0.419) between type I and type II os trigonum, respectively. These results are depicted graphically in Figure 2.

#### 3.2.2. Relationship between Type and Magnetic Resonance Imaging Findings

In the group of patients with os trigonum, detachment status was recorded in terms of scale 0 (33/36, 91.7%), scale 1 (2/36, 5.6%), and scale 2 (1/36, 2.8%) with type I and scale 0 (51/88, 58.0%), scale 1 (21/88, 23.9%) and scale 2 (16/88, 18.2) with type II. The results showed that the MR image of a patient with a type I os trigonum tends to have less detachment of the os trigonum from the posterior talar cortex than type II (91.7% vs. 58.0%). More frequent detachment was visible in the type II os trigonum patient than in type I (23.9% and 18.2% vs. 5.6.% and 2.8% in each scale 1 and 2). Differences between the groups were statistically significant (*p*  =  0.001).

In patients with type I os trigonum, PTFL scale 0 was found in 6/36 (16.6%), and scale 1 was found in 30/36 (83.4%). PTFL attachment scale 0 was not found in the type I group. In patients of type II os trigonum, scale 0 was found in 11/88 (12.5%), scale 1 was found in 65/88 (%), and scale 2 was found in 12/88 (13.6%). PTFL attachment scale 2 was found in only type II os trigonum, but no significant statistical differences existed between type I and II os trigonum (*p* = 0.63).

In terms of the PTFL abnormality, a high-grade abnormality that represents the discontinuity of PTFL (grade 2) was more frequently found in the case of type II (47/88, 53.4%) than type I (3/36, 8.3%) but this did not meet statistical significance. There was no statistically significant variance between type I and II os trigonum in the scale of the PTFL abnormality (*p*  =  0.708), bone marrow edema in os trigonum (*p*  =  0.981), FHL tenosynovitis (*p*  =  0.943), or posterior synovitis (*p*  =  0.390). Detailed results are listed in Table 2.

#### 3.2.3. Inter-Observer Agreement

There was almost perfect agreement between readers in determining the type of os trigonum (κ = 0.98) and substantial agreement for PTFL attachment (κ = 0.72), PTFL abnormality (κ = 0.76), bone marrow edema (κ = 0.67), detachment (κ = 0.80), and posterior synovitis (κ = 0.71). The inter-observer agreement for the FHL tenosynovitis was moderate (κ = 0.48) (Table 3). Overall, the concordance rate of type and each radiologic finding was 62.9% (FHL tenosynovitis) to 99.2% (type of os trigonum).

## 4. Discussion

The os trigonum is a small accessory bone located posterior to the talus in the ankle joint. It is present in approximately 7–14% of individuals. It is believed to arise from the failure of fusion between the lateral tubercle of the posterior process of the talus and the surrounding bone during development [7]. The os trigonum can cause pain and limited range of motion in the ankle joint, particularly in athletes and dancers who place increased stress on this area [8]. MRI is commonly used to diagnose and evaluate os trigonum, as it allows for visualization of the bony structure and surrounding soft tissues.

Grossly, os trigonum appears as a small, round, irregularly shaped bone posterior to the talus. It may be smooth or rough, and its size and shape can vary between individuals. Microscopically, os trigonum is composed of normal bone tissue, with no significant differences in histological features compared to other bones in the body. Treatment usually begins with nonsurgical measures, including physical therapy. However, in cases where os trigonum is causing symptoms, surgical excision is often recommended to relieve pain and restore function [9]. Additionally, arthroscopic treatment has been reported for os trigonum syndrome with concomitant posterolateral osteochondral lesion of the talus, resulting in improved symptoms and healing [10].

The classification of os trigonum refers to categorizing the os trigonum bone based on its size, shape, and appearance in imaging studies [11]. Different classification systems describe os trigonum, but one commonly used system divides it into three types. Zwiers et al. [5] classified the os trigonum into three types based on its appearance on X-ray imaging. Type I os trigonum appeared as a separate ossicle with the talar tubercle in its usual appearance. Type II os trigonum was characterized by the ossicle being part of the talar tubercle. Finally, type III os trigonum was identified as an ossicle that had developed in the area without the development of the talar tubercle. This classification system allowed for a more detailed understanding of the different types of os trigonum and their associated features. It may have important implications for diagnosing and treating os trigonum-related pathologies, such as posterior ankle impingement syndrome. In our study, to investigate the pathology of the FHL tendon and its relationship with the os trigonum, we categorized it into two types, according to whether the os trigonum invaded the FHL groove.

Os trigonum can contribute to the development of posterior ankle impingement syndrome. This syndrome is characterized by posterior ankle pain with forceful plantar flexion, and the os trigonum, a secondary ossification center at the posterolateral corner of the talus, is the most common cause of posterior impingement syndrome [12]. The presence of os trigonum can increase the likelihood of impingement because it can impinge on the soft tissues and bone structures in the ankle joint, leading to inflammation, swelling, and pain. The syndrome can be acute or chronic, with symptoms exacerbated by forced plantar flexion or wearing high-heeled shoes [13]. In addition, os trigonum can create a secondary ossicle–tendon syndrome, where the ossicle rubs against the flexor hallucis longus (FHL) tendon, causing pain and discomfort.

According to our hypothesis, the severity of posterior ankle impingement syndrome may depend on the type of os trigonum present. A type 2 os trigonum, which is larger and more irregularly shaped than type 1, has been associated with a higher risk of posterior ankle impingement syndrome. In our study, MRI findings of type II os trigonum, compared to type I, showed a higher frequency of PTFL discontinuity (8.3% vs. 13.6%) and severe posterior synovitis (58.3% vs. 68.2%). These results suggest that type II os trigonum may have a more significant impact on posterior ankle pain and impingement than type I, but this was not statistically significant.

Our study’s results demonstrate a significant relationship between the type of os trigonum and the degree of detachment from the posterior talar cortex seen on MRI. We found that patients with type I os trigonum demonstrated less detachment compared to those with type II os trigonum (91.7% vs. 58.0%), while type II os trigonum showed more frequent detachment (23.9% and 18.2% in each of grade 1 and 2, respectively) than type I os trigonum (5.6% and 2.8% in each of grade 1 and 2, respectively). These findings suggest that the morphology of the os trigonum may play a significant role in the development of posterior ankle impingement syndrome. Additionally, we found that PTFL abnormality, including the thickening, thinning, or discontinuity of PTFL, was more frequently found in patients with type II os trigonum (46.6%) than in patients with type I os trigonum (44.3%). Although this difference did not reach statistical significance, it suggests that patients with type II os trigonum may be at a higher risk of developing PTFL abnormalities. The study also examined other MRI findings, such as bone marrow edema in os trigonum, FHL tenosynovitis, and posterior synovitis. However, no significant differences were found between type I and type II os trigonum.

The technical parameters of MRI, such as field strengths and slice thicknesses, have a significant impact on the evaluation of os trigonum types and their anatomical relationships. Higher field strength MRI scanners typically offer a better signal-to-noise ratio (SNR) and spatial resolution, which enhances the delineation of small structures like the os trigonum. This improved resolution is crucial in accurately classifying the os trigonum types and in assessing their relationship with adjacent structures. However, variations in field strength can also lead to differences in contrast resolution and susceptibility artifacts, which might affect the visualization of certain tissue types and interfaces.

Regarding slice thickness, thinner slices are generally advantageous for detailed anatomical evaluations, particularly for small ossicles like the os trigonum. They reduce partial volume effects, thereby allowing for more precise measurement of the os trigonum’s dimensions and a clearer depiction of its morphology. On the other hand, thicker slices, while reducing scan time, might compromise the ability to detect subtle variations in size and shape, which are crucial for classifying os trigonum types and understanding their biomechanical implications. Thus, while our study aimed to standardize these parameters as much as possible, inherent variations due to different MRI scanners may have influenced the determination of os trigonum types and measurements.

In the evaluation of ankle pathologies, particularly os trigonum, it is imperative to also consider the differential diagnosis of Shepherd’s and Cedell fractures. Os trigonum syndrome and Shepherd’s fractures both involve the posterior aspect of the talus but differ significantly in their etiology and radiological appearance. Radiologically, an os trigonum appears as a well-corticated ossicle adjacent to the posterior talar process on lateral ankle X-rays. In contrast, a Shepherd’s fracture, a type of avulsion fracture, results from acute trauma, typically an ankle sprain. This fracture is characterized by a fragment at the lateral tubercle of the posterior process of the talus [14]. A Cedell fracture, a less common variant, involves the posteromedial tubercle and is often associated with more significant trauma [15]. The diagnosis of these fractures relies on detecting irregularities or discontinuities in the cortical line of the posterior talus on lateral or oblique radiographs, and sometimes necessitates CT or MRI for confirmation.

Differential diagnosis between these conditions is critical for appropriate management. In os trigonum syndrome, conservative treatment with rest, ice, and physical therapy is often effective, but surgical intervention may be required for persistent symptoms. For Shepherd’s and Cedell fractures, treatment depends on the displacement of the fracture fragment and the stability of the ankle joint. Non-displaced fractures are typically managed conservatively with immobilization and physiotherapy, while displaced fractures may require surgical fixation. Accurate diagnosis through careful radiological assessment is essential to guide these treatment decisions and ensure optimal patient outcomes.

In some cases, surgical excision of the os trigonum may be necessary to alleviate symptoms and restore normal ankle function. Clinicians should consider the os trigonum type when diagnosing and treating patients with posterior ankle pain. So, the relationship between the type of os trigonum and posterior ankle impingement syndrome highlights the importance of accurate diagnosis and management of os trigonum in individuals with posterior ankle pain.

The limitations of this study include that our study was conducted at a single center with a relatively small patient sample size. Due to the rarity of os trigonum, recruiting a large sample of patients with this condition can be challenging. Furthermore, the study relied on retrospective analysis of MRI images, which may have introduced bias in interpreting the results. The lack of outside observers in the study may have also affected the validity of the findings. In future studies, it would be beneficial to conduct a multi-center study with a larger sample size to increase the generalizability of the results. Involving outside observers in interpreting MRI images also helps reduce the potential for bias. Another limitation of this study is that there may be heterogeneity in the MRI data due to the use of multiple scanners with varying image quality parameters such as resolution, contrast, and signal-to-noise ratio (SNR). This is because our study design assumed that images from all MRI scanners in the hospital would be utilized, which is the same as in a real-world clinical setting, and we will consider taking this aspect into account in future studies to further validate our results.

## 5. Conclusions

Os trigonum is a small accessory bone that can contribute to the development of posterior ankle impingement syndrome. In our study, the severity of posterior ankle impingement syndrome may depend on the type of os trigonum present. Type II os trigonum is associated with a higher risk of impingement and PTFL abnormalities than type I os trigonum. The results of our study demonstrate the importance of accurate diagnosis and management of os trigonum in individuals with posterior ankle pain.

In summary, the relationship between os trigonum and posterior ankle impingement syndrome is complex, requiring a thorough understanding of the different types of os trigonum and their associated features. Clinicians should consider the os trigonum type when diagnosing and treating patients with posterior ankle pain, and surgical excision may be necessary in some cases to alleviate symptoms and restore normal ankle function. Further research is needed to validate these findings and determine the most effective treatment strategies for os trigonum-related pathologies.

## Figures and Tables

**Figure 1 diagnostics-14-00283-f001:**
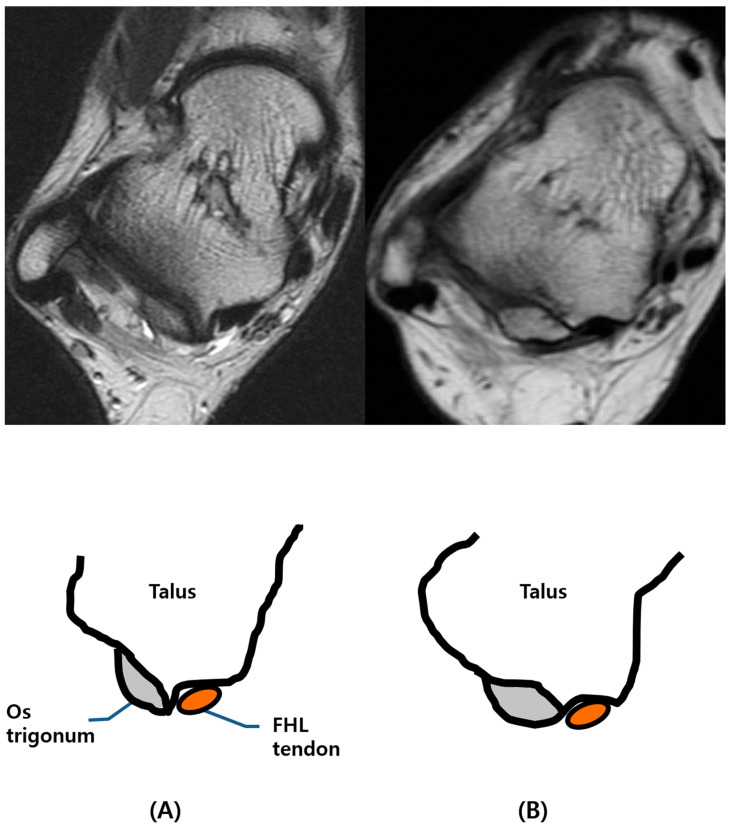
Different types of os trigonum. (**A**) Type I os trigonum with intact FHL groove, (**B**) type II os trigonum as the medial border of the ossicle-invading groove for FHL tendon.

**Figure 2 diagnostics-14-00283-f002:**
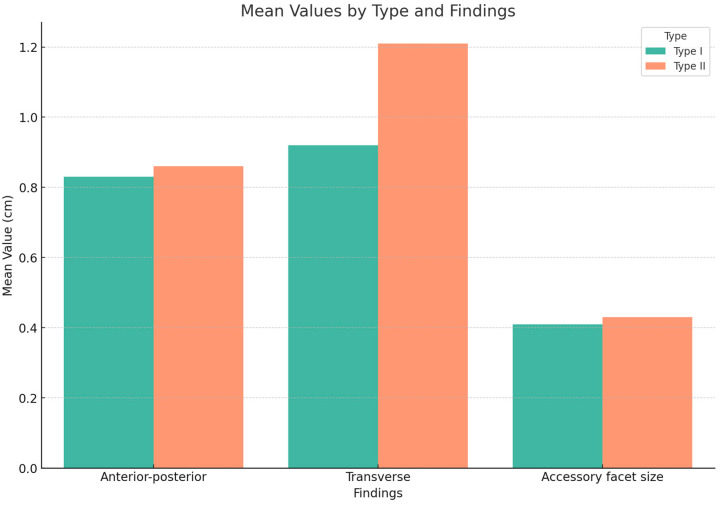
Dimensions of the os trigonum in each type.

**Table 1 diagnostics-14-00283-t001:** Demographics and type of the os trigonum.

Characteristic	Value		
Number of Patients (Cases)	123 (124)		
Type of os trigonum (*n* = 124)	Type I	Type II	*p* value
	36 (29.0%)	88 (71.0%)	
Age (years)			
Mean age (± SD)	30.3 (±10.77)	37.1 (±15.5)	0.018 *
Sex (*n* = 123)			
Male	27 (75.0%)	56 (64.3%)	0.298
Female	9 (25.0%)	31 (35.7%)	
Count of ossicles (*n* = 124)			
1	35 (97.2%)	83 (94.3%)	0.671
2	1 (2.8%)	5 (5.7%)	

* Indicates a *p*-value of less than 0.05, denoting statistical significance.

**Table 2 diagnostics-14-00283-t002:** MR imaging findings and scale distribution of type I and II os trigonum.

Variables			Type		Χ^2^/*p*
Findings	Grade	Description	Type I	Type II	
PTFL attachment	0	Normal	6 (16.6%)	11 (12.5%)	5.532/0.63
	1	Attachment < 1/2	30 (83.4%)	65 (73.9%)	
	2	≥1/2	0 (0%)	12 (13.6%)	
PTFL abnormality	0	Normal	20 (55.6%)	47 (53.4%)	0.691/0.708
	1	Thickening or thinning	13 (36.1%)	29 (33.0%)	
	2	Discontinuity	3 (8.3%)	12 (13.6%)	
Bone marrow edema	0	No BME	10 (27.8%)	26 (29.5%)	0.39/0.981
	1	<1/2 of os	21 (58.3%)	50 (56.8%)	
	2	≥1/2 of os	5 (13.9%)	12 (13.6%)	
Detachment	0	No detachment	33 (91.7%)	51 (58.0%)	13.325/0.001 *
	1	Partial detachment	2 (5.6%)	21 (23.9%)	
	2	Complete detachment	1 (2.8%)	16 (18.2%)	
FHL tenosynovitis	0	Normal	3 (8.3%)	9 (10.2%)	0.117/0.943
	1	Mild tenosynovitis	18 (50.0%)	44 (50.0%)	
	2	Severe tenosynovitis	15 (41.7%)	35 (39.8%)	
Posterior synovitis	0	Normal	3 (8.4%)	9 (10.2%)	1.883/0.390
	1	Mild synovitis	12 (33.3%)	19 (21.6%)	
	2	Severe synovitis	21 (58.3%)	60 (68.2%)	
Total			36 (29.0%)	88 (71.0%)	

* Indicates a *p*-value of less than 0.05, denoting statistical significance.

**Table 3 diagnostics-14-00283-t003:** Inter-observer agreement and concordance rate in the independent review session.

Feature	K Coefficient	Concordance Rate (%) *
Type of os trigonum	0.98	99.2% (123/124)
PTFL attachment	0.72	87.9% (109/124)
PTFL abnormality	0.76	79.8% (99/124)
FHL tenosynovitis	0.48	62.9% (78/124)
BME	0.67	76.6% (95/124)
Detachment	0.80	87.1% (108/124)
Posterior synovitis	0.71	82.3% (102/124)

* Data are the percentage of times readers provided concordant results for the same patient for the same imaging feature; numbers in parentheses are numbers of patients.

## Data Availability

The data presented in this study are available on request from the corresponding author. The data are not publicly available due to need approval from the affiliated institution’s DRB (Data review board) is required for disclosure or export.

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
