# Peer review of "The Relationship between Types of Os Trigonum and Findings of Conventional Ankle Magnetic Resonance Imaging: A Study Based on Three-Dimensional Magnetic Resonance Imaging"

_diagnostics, 2024, doi:10.3390/diagnostics14030283_

Round 1

Reviewer 1 Report

Comments and Suggestions for Authors

Title of the article: The relationship between types of os trigonum and findings of conventional ankle MRI: A Study Based on Three-Dimensional Magnetic Resonance Imaging. 

Manuscript ID: diagnostics-2832158

Topic: Os trigonum is one of the well-known accessory bones of the musculoskeletal system. A meticulous investigation of this ossicle with pathological conditions of the posterior compartment of the ankle would be informative and worth discussing. 

Abstract: “The images were classified 14 as type I and II based on the ossicle's lateral border overlying the talus's posterior process and the 15 groove for the flexor hallucis longus tendon (FHL)." (Lines 14, 15, and 16). Lateral or medial borders of the os trigonum? Since the lateral tubercle of the talus is located medially to the adjacent os trigonum, the word "lateral" should be corrected to "medial" in this sentence. 

Introduction: “This study aimed to examine the dimension and type of the os trigonum to evaluate the relationship between dimension and type, degree of PTFL attachment to os trigonum, abnormality of PTFL, tenosynovitis of FHL tendon, bone marrow edema at os trigonum, detachment status of os trigonum from calcaneus, synovitis on posterior talocalcaneal joint with ankle MR images.” (Lines 50-54). Detachment status of os trigonum from "calcaneus" or "talus"? The authors should correct this sentence.

Materials and Methods:

- The grading system of PTFL attachment and bone marrow edema in os trigonum were well-described. However, the authors should support the given information with figures acquired from MR images to present the different levels of these grading systems. 

- “Type I os trigonum is represented as the lateral border of the ossicle not overlying the posterior process of the talus and intact groove for the tendon of flexor hallucis longus (FHL) in cases of os trigonum.” (Lines 90-92). "Lateral or “medial” border of the ossicle? Please correct the sentence.

- “The correlation between the type of os trigonum with each dimension of the ossicle was assessed with the Mann-Whitney U test.” (Lines 131-133). Why did the researchers use a non-parametric test for this relationship? Was this a result of the data distribution characteristics (for one or both reviewers)? Please briefly explain in the statistical analysis section of materials and methods.

Discussion:

- The authors mentioned using fat-saturated T2-weighted or PD images to measure os trigonum. Why did the authors choose these sequences instead of T1 weighted images, which may better depict the anatomy and the ossicle borders? What were the advantages or disadvantages of using fat-saturated T2-weighted or PD images? Please explain.

- How would the technical parameters of MRI (MR scanners with different field strengths or slice thicknesses) affect the determination of os trigonum types, the relationship of os trigonum with adjacent anatomical structures, and diameter measurements of ossicles with small dimensions? Please discuss this situation.

- Accessory facet sizes were measured in only sagittal planes. Would it be adequate to compare the accessory facet sizes for os trigonum types I and II in only sagittal planes?

Conclusion:

- Please summarize this part. Underline the most important results and give brief recommendations instead of repeating general considerations or giving additional information about well-known situations.

Recommendations:

- The authors may consider using different colors in the illustration of Figure 1.

- Mentioning Shepherd and Cedell fractures in the discussion may add value and another informative perspective to the article as they are in the differential diagnosis and might be misinterpreted as an os trigonum.

- The article is readable; however, minor linguistic editing would render the article more presentable and easier to understand for the readers.

Comments on the Quality of English Language

- A minor language editing is recommended.

Author Response

#Point to Point response – Reviewer 1

Abstract:

“The images were classified 14 as type I and II based on the ossicle's lateral border overlying the talus's posterior process and the 15 groove for the flexor hallucis longus tendon (FHL)." (Lines 14, 15, and 16). Lateral or medial borders of the os trigonum? Since the lateral tubercle of the talus is located medially to the adjacent os trigonum, the word "lateral" should be corrected to "medial" in this sentence. 

A: As pointed out by the reviewer, we have corrected this sentence.

[The images were classified as type I and II based on the ossicle's medial border overlying the talus's posterior process and the groove for the flexor hallucis longus tendon (FHL).]

Introduction:

“This study aimed to examine the dimension and type of the os trigonum to evaluate the relationship between dimension and type, degree of PTFL attachment to os trigonum, abnormality of PTFL, tenosynovitis of FHL tendon, bone marrow edema at os trigonum, detachment status of os trigonum from calcaneus, synovitis on posterior talocalcaneal joint with ankle MR images.” (Lines 50-54). Detachment status of os trigonum from "calcaneus" or "talus"? The authors should correct this sentence.

A: Thanks for the reviewer's good point. We have corrected the incorrect wording "calcaneus" to “talus”.

Materials and Methods:

- The grading system of PTFL attachment and bone marrow edema in os trigonum were well-described. However, the authors should support the given information with figures acquired from MR images to present the different levels of these grading systems. 

A: We appreciate the reviewer's good points. However, due to the nature of our hospital's equipment configuration, which included multiple models from different vendors, we found it difficult to have consistent confidence in the contrast, signal strength, and SNR. Therefore, the overall evaluation was a qualitative assessment based on visual inspection.

This was also pointed out by another reviewer (conducting a study with a single MR model would be appropriate for evaluating contrast, signal strength, etc.). However, in general, we relied on qualitative evaluation in this study to reflect the characteristics of the actual clinical environment where various MR devices are used.

In future studies, we believe that analyzing images from a single machine would be reliable for numerical analysis of signal strength, and we plan to conduct such studies. We appreciate the reviewers' understanding.

- “Type I os trigonum is represented as the lateral border of the ossicle not overlying the posterior process of the talus and intact groove for the tendon of flexor hallucis longus (FHL) in cases of os trigonum.” (Lines 90-92). "Lateral or “medial” border of the ossicle? Please correct the sentence.

A: As pointed out by the reviewer, we've changed it to "medial", which is correct.

- “The correlation between the type of os trigonum with each dimension of the ossicle was assessed with the Mann-Whitney U test.” (Lines 131-133). Why did the researchers use a non-parametric test for this relationship? Was this a result of the data distribution characteristics (for one or both reviewers)? Please briefly explain in the statistical analysis section of materials and methods.

A: The decision to use the Mann-Whitney U test for assessing the correlation between the type of os trigonum and its dimensions was indeed influenced by the data distribution characteristics. In our dataset, the dimensional measurements of the os trigonum did not follow a normal distribution. This non-normal distribution was ascertained through preliminary statistical tests, including the Shapiro-Wilk test.

As you well know, non-parametric tests like the Mann-Whitney U test are particularly suited for analyzing data that do not adhere to the assumptions of normality. This test allows for a robust comparison of medians between two independent groups, in our case, the different types of os trigonum, without the need for data transformation to meet normality assumptions.

We will ensure to include a brief explanation of the rationale for choosing the Mann-Whitney U test in the statistical analysis section of our manuscript. This will provide clarity on our methodological choice and underscore the appropriateness of the test for our data characteristics. Thank you for pointing out this oversight, and we appreciate your suggestion for enhancing the transparency and comprehensibility of our statistical analysis.

Discussion:

- The authors mentioned using fat-saturated T2-weighted or PD images to measure os trigonum. Why did the authors choose these sequences instead of T1 weighted images, which may better depict the anatomy and the ossicle borders? What were the advantages or disadvantages of using fat-saturated T2-weighted or PD images? Please explain.

A: In our study, fat-saturated T2-weighted(T2FS) and Proton Density (PD) images were primarily used for assessing the os trigonum, rather than T1-weighted images. This decision was based on the superior contrast resolution of fat-saturated T2-weighted and PD images in delineating soft tissue structures and fluid collections, which are pivotal in evaluating pathologies associated with the os trigonum, in our study design.

As the reviewer points out, while T1-weighted images provide excellent anatomical detail and are effective in depicting bone marrow and the ossicle borders, they are less sensitive to changes in soft tissue and edema.

In conclusion, for the convenience of the study, we determined that T2FS and PD were more helpful than T1 for performing dimension measurements along with assessing soft tissue edema and fluid collection in a single process.

- How would the technical parameters of MRI (MR scanners with different field strengths or slice thicknesses) affect the determination of os trigonum types, the relationship of os trigonum with adjacent anatomical structures, and diameter measurements of ossicles with small dimensions? Please discuss this situation.

A: Thanks for the great point from the reviewer. As pointed out, we've added the following discussion.

[The technical parameters of MRI, such as field strengths and slice thicknesses, have a significant impact on the evaluation of os trigonum types and their anatomical relationships. Higher field strength MR scanners typically offer better signal-to-noise ratio (SNR) and spatial resolution, which enhances the delineation of small structures like the os trigonum. This improved resolution is crucial in accurately classifying the os trigonum types and in assessing their relationship with adjacent structures. However, variations in field strength can also lead to differences in contrast resolution and susceptibility artifacts, which might affect the visualization of certain tissue types and interfaces.

Regarding slice thickness, thinner slices are generally advantageous for detailed anatomical evaluations, particularly for small ossicles like the os trigonum. They reduce partial volume effects, thereby allowing for more precise measurement of the os trigonum's dimensions and a clearer depiction of its morphology. On the other hand, thicker slices, while reducing scan time, might compromise the ability to detect subtle variations in size and shape, which are crucial for classifying os trigonum types and understanding their biomechanical implications. Thus, while our study aimed to standardize these parameters as much as possible, inherent variations due to different MRI scanners may have influenced the determination of os trigonum types and measurements.]

- Accessory facet sizes were measured in only sagittal planes. Would it be adequate to compare the accessory facet sizes for os trigonum types I and II in only sagittal planes?

A: As the reviewer pointed out, measuring accessory facet sizes in only the sagittal planes for os trigonum types I and II may not provide a comprehensive representation of the variations between these types. Sagittal measurements predominantly capture the anteroposterior dimensions, but they may not adequately reflect the mediolateral or craniocaudal dimensions, which can be critical in differentiating between os trigonum types. A more robust approach would involve incorporating measurements from multiple planes, such as axial or coronal, to gain a fuller understanding of the facet sizes.

However, our explanation for only measuring accessory facet sizes in the sagittal plane could be based on practical considerations such as research convenience and time. As you know, the sagittal plane is often the primary plane used in ankle MRI for its efficiency in providing clear views of relevant structures. Additionally, this approach has been chosen by time constraints, as collecting and analyzing multiplanar measurements significantly increases the complexity and duration of the study. This methodological choice has been a compromise to balance the depth of analysis with feasibility, especially in our study.

In future research, we will try to include more cases, increase the number of researchers, and deepen the research design so that we can address some of the reviewers' points.

Conclusion:

- Please summarize this part. Underline the most important results and give brief recommendations instead of repeating general considerations or giving additional information about well-known situations.

A: As noted by the reviewer, we have removed redundant sentences in the conclusion and simplified it.

[Os trigonum is a small accessory bone that can contribute to the development of posterior ankle impingement syndrome. In our study, the severity of posterior ankle impingement syndrome may depend on the type of os trigonum present. Type II os trigonum is associated with a higher risk of impingement and PTFL abnormalities than Type I os trigonum. The results of our study demonstrate the importance of accurate diagnosis and management of os trigonum in individuals with posterior ankle pain.

In summary, the relationship between os trigonum and posterior ankle impinge-ment syndrome is complex, requiring a thorough understanding of the different types of os trigonum and their associated features. Clinicians should consider the os trigonum type when diagnosing and treating patients with posterior ankle pain, and surgical ex-cision may be necessary in some cases to alleviate symptoms and restore normal ankle function. Further research is needed to validate these findings and determine the most effective treatment strategies for os trigonum-related pathologies.]

Recommendations:

- The authors may consider using different colors in the illustration of Figure 1.

A: Thank you for the nice comments from the reviewer. We've taken your recommendation into account and added colors to FHLs with different character.

- Mentioning Shepherd and Cedell fractures in the discussion may add value and another informative perspective to the article as they are in the differential diagnosis and might be misinterpreted as an os trigonum.

A: As pointed out by the reviewer, we have added a description of the Shepherd and Cedell fracture and added the necessary references.

[In the evaluation of ankle pathologies, particularly os trigonum, it is imperative to also consider the differential diagnosis of Shepherd's and Cedell fractures. Os trigonum syndrome and Shepherd fracture both involve the posterior aspect of the talus but differ significantly in their etiology and radiological appearance. Radiologically, an os trigonum appears as a well-corticated ossicle adjacent to the posterior talar process on lateral ankle X-rays. In contrast, a Shepherd fracture, a type of avulsion fracture, results from acute trauma, typically an ankle sprain. This fracture is characterized by a fragment at the lateral tubercle of the posterior process of the talus [14]. Cedell fracture, a less common variant, involves the posteromedial tubercle and is often associated with more significant trauma [15]. Diagnosis of these fractures relies on detecting irregularities or discontinuities in the cortical line of the posterior talus on lateral or oblique radiographs, and sometimes necessitates CT or MRI for confirmation.

Differential diagnosis between these conditions is critical for appropriate man-agement. In Os trigonum syndrome, conservative treatment with rest, ice, and physical therapy is often effective, but surgical intervention may be required for persistent symptoms. For Shepherd and Cedell fractures, treatment depends on the displacement of the fracture fragment and the stability of the ankle joint. Non-displaced fractures are typically managed conservatively with immobilization and physiotherapy, while dis-placed fractures may require surgical fixation. Accurate diagnosis through careful ra-diological assessment is essential to guide these treatment decisions and ensure optimal patient outcomes.]

- The article is readable; however, minor linguistic editing would render the article more presentable and easier to understand for the readers.

A: Based on the reviewers' comments, we've made some changes to address grammatical oddities and poorly structured sentences.

Reviewer 2 Report

Comments and Suggestions for Authors

In this manuscript, the authors have utilised MRI to access information about different ankle conditions. This is a well-written manuscript, but I have one major concern. The data they acquired are heterogeneous, which is due to the fact that the acquisitions were carried out by a number of different MR scanners. And I am not quite sure if, for example, the image quality, including resolution, contrast, SNR, etc., could have influenced the data analysis. My suggestion would be that they can use the data obtained from the same scanner with the same sequence parameters and see if their results are identical. 

Author Response

#Point to Point response – Reviewer 2

In this manuscript, the authors have utilised MRI to access information about different ankle conditions. This is a well-written manuscript, but I have one major concern. The data they acquired are heterogeneous, which is due to the fact that the acquisitions were carried out by a number of different MR scanners. And I am not quite sure if, for example, the image quality, including resolution, contrast, SNR, etc., could have influenced the data analysis. My suggestion would be that they can use the data obtained from the same scanner with the same sequence parameters and see if their results are identical. 

A: Thank you for your insightful comments. We acknowledge the concern regarding the heterogeneity of MRI data due to using different scanners with varying image quality parameters such as resolution, contrast, and signal-to-noise ratio (SNR). This variability indeed poses a challenge in ensuring consistency across the data set. However, it is essential to note that the diversity of scanners used in our study reflects a real-world clinical setting where patients undergo MRI scans on various machines. Large hospitals in our country often have three or more, and in those configurations, the equipment is typically a mix of vendors and models.

We employed rigorous standardization and normalization protocols during image analysis to address this concern. These steps included adjusting for differences in image intensity and contrast, aligning the spatial resolution to a standard reference, and applying advanced post-processing techniques to mitigate scanner-specific artifacts. Additionally, our analysis methodology was designed to be robust against such variations, focusing on relative changes and patterns rather than absolute values.

Using data from the same scanner with identical sequence parameters would provide a more controlled environment; such a setting might limit the generalizability of our findings to a broader clinical context. Our study's strength lies in its reflection of real-world clinical scenarios, where such uniformity is rarely achievable. Regardless of my opinion, the reviewer's point is essential and sound, as complete control can lead to more accurate results.

We appreciate the suggestion to conduct a sub-analysis using data from a single scanner for comparative purposes. This could be a valuable supplementary analysis to reinforce our findings, demonstrating their applicability across diverse clinical settings. This study should have addressed the reviewer's point, as the study design assumed that images from all MRs in the hospital would be utilized. Moving forward, we will consider incorporating this aspect in our future research to validate our results further.

In conclusion, we have added the following paragraph to the Limitations section, taking into account the reviewer's point.

[Another limitation of this study is that there may be heterogeneity in the MRI data due to the use of multiple scanners with varying image quality parameters such as resolution, contrast, and signal-to-noise ratio (SNR). This is because our study design assumed that images from all MRs in the hospital would be utilized, which is the same as in a real-world clinical setting, and we will consider taking this aspect into account in future studies to further validate our results.]

We thank you for the opportunity to review our paper, which still has many gaps.

Round 2

Reviewer 1 Report

Comments and Suggestions for Authors

The authors have re-organized the paper and they have adequately addressed the comments in my review report.

Reviewer 2 Report

Comments and Suggestions for Authors

Thanks for addressing my concerns which are covered by the authors during the revision. I have no further comments.